# Online Model Adaptation
# with Feedforward Compensation

**Abulikemu Abuduweili,    Changliu Liu**
Robotics Institute, Carnegie Mellon University
{abulikea,cliu6}@andrew.cmu.edu

**Abstract:** To cope with distribution shifts or non-stationarity in system dynamics, online adaptation algorithms have been introduced to update offline-learned prediction models in real-time. Existing online adaptation methods focus on optimizing the prediction model by utilizing feedback from the latest prediction error. However, this feedback-based approach is susceptible to forgetting past information. This work proposes an online adaptation method with feedforward compensation, which uses critical data samples from a memory buffer, instead of the latest samples, to optimize the prediction model. We prove that the proposed approach achieves a smaller error bound compared to previously utilized methods in slow time-varying systems. Furthermore, our feedforward adaptation technique is capable of estimating an uncertainty bound for predictions.

**Keywords:** Online Adaptation, Optimization, Behavior Prediction

## 1 Introduction

Robots rely on prediction models of human behavior or other agents to plan safe and efficient motions within their environment. The development or learning of accurate and reliable behavior prediction models is crucial for the successful deployment of autonomous systems in environments involving humans or diverse agents [1, 2, 3]. However, creating a universal prediction model or algorithm capable of accurately predicting different agents in various scenarios is extremely challenging. Recognizing this limitation, it becomes necessary and advantageous for robots to have the capability to adapt their prediction models online when a high-fidelity behavior prediction model is not available [4, 5, 6, 7]. For instance, consider the prediction of human motion in human-robot collaboration, due to variations in individual preferences and behavioral styles, prediction models that effectively capture one person's behavior may not be applicable to another individual. In such cases, online adaptation of the human motion prediction model becomes essential, allowing the robot to adapt its predictions by observing the specific behaviors of the individual [8, 9]. Similarly, in the context of autonomous driving, the prediction of surrounding road participants' behavior necessitates adaptation [10, 11]. Driving behavior is subject to change based on road conditions, weather, and various environmental factors. During the training phase, it is impractical to collect data encompassing all potential scenarios. Consequently, the prediction model of the agent must adapt online while observing real-time driving behaviors.

In online adaptation, a prediction model receives sequential observations, and an online optimization algorithm (such as stochastic gradient descent, SGD) is employed to update the model based on the prediction loss computed from the observed data [12]. The goal of online adaptation is to improve prediction accuracy in subsequent rounds. Most existing online adaptation approaches are based on feedback compensation [13, 14, 15], analogous to feedback control. In feedback adaptation, the prediction model exclusively incorporates the most recent data received. Following the observation of a new sample, the online optimization algorithm updates the prediction model by calculating the prediction loss between the previous prediction and the latest ground truth. However, this feedback compensation approach is prone to disregarding past information, thus limiting its effectiveness.

7th Conference on Robot Learning (CoRL 2023), Atlanta, USA.

In this work, we propose feedforward compensation in online adaptation to maximize information extraction from existing data, especially those that are more important. In the proposed **feedforward adaptation**, we enable recalling to compensate for potential shortsighted behaviors due to forgetting in general online adaptation [16]. To achieve a balance between forgetting and recalling, we maintain a memory buffer by storing recent $L$-steps observations. When similar trajectories are observed, the feedforward adaptation method will pull similar samples from the memory buffer to enhance learning. For example, in human behavior prediction tasks, a human subject may exhibit similar behavior patterns on different days. Discovering such patterns becomes challenging using conventional online adaptation that solely relies on the most recent data, whereas our feedforward adaptation method with a memory buffer facilitates their identification. In addition, the proposed feedforward adaptation is capable of estimating uncertainty bound by comparing the current observation and historical observations. The proposed approach shares similarity with replay-based continual learning [17], as both methods involve retaining critical past samples in a buffer and subsequently replaying them to enhance the learning process. However, many replay-based continual learning techniques employ random replay mechanisms [18, 19] or focus on task or class incremental problems [17, 20]. Furthermore, online adaptation emphasizes local performance, while continual learning emphasizes generality. Further discussion on this topic is available in Appendix G. Our main contributions are summarized below[1].

- We introduce a general online adaptation framework.
- We propose a feedforward compensation method in online adaptation. The proposed feedforward method has a smaller error bound than feedback methods in slow time-varying systems.
- We propose a method for uncertainty-bound estimation, which is agnostic from optimizers.
- We conduct extensive experiments to show that the proposed feedforward adaptation is superior to conventional feedback adaptation.

## 2   Problem Overview

**Behavior prediction**    Behavior prediction is a sub-topic of time series prediction, which mainly focuses on predicting future motions (e.g. trajectory) of human or other agents given the past and current observations [21, 22, 23]. Assume the transition function of agent's behavior is denoted as $f$. At time step $t$, the input to the function is $X_t = [x_{t-I+1}, x_{t-I+2}, \cdots, x_t]$, which denotes the stack of $I$-step recent observations (or states). The output of the function is $Y_{t+1} = [y_{t+1}, y_{t+2}, \cdots, y_{t+O}]$, which denotes the stack of $O$-step future predictions. The observations $x_t$ and $y_t$ are vectors that may contain trajectories or features, and $x_t = y_t$ for some cases (e.g. univariate prediction).

$$Y_{t+1} = f(t, X_t). \tag{1}$$

For prediction, we use a parameterized model (e.g. Neural Networks) $\hat{f}(\theta_t, X_t)$ with learnable parameters $\theta_t$ to estimate the ground-truth transition function $f(t, X_t)$. The analysis of the online adaptation in the paper is based on the following two conditions about the local smoothness of transition function $f$. Assume we maintain a $L$-size buffer to store recent $L$-steps historical observations.

**Assumption 1.** $K$-*Lipschitz continuity condition. At time step $t$, the function $f$ is locally $K$ Lipschitz continuious for the recent $L$ steps input data, if the following holds $\forall s \in [t - L, t - 1]$:*

$$\|f(t, X_t) - f(t, X_s)\| \le K\|X_t - X_s\|, \tag{2}$$

where $K$ is the bound (real number) for the change of the value of the function over input (observation) space. Intuitively, a Lipschitz continuous function is limited in how dramatically the function value can change over input space. Similar to (2), we assume our parameterized function $\hat{f}_t(\theta_t, :)$ (e.g. Neural Networks) is locally Lipschitz continuous $\forall s \in [t - L, t - 1]$, with constant value $\hat{K}$:

$$\|\hat{f}(\theta_t, X_s) - \hat{f}(\theta_t, X_t)\| \le \hat{K}\|X_t - X_s\|. \tag{3}$$

---

[1]The code of the paper is available at https://github.com/intelligent-control-lab/Feedforward_Adaptation.

**Assumption 2.** $\delta$ **time-varying condition.** *At time step $t$, a transition function $f$ is $\delta$ time-varying for the recent $L$ steps under the input $X_s$, if the following holds $\forall s \in [t - L, t - 1]$:*

$$\|f(t, X_s) - f(s, X_s)\| \leq \delta |t - s|, \tag{4}$$

where $\delta \in \mathbb{R}^+$ is the bound for the change rate on a fixed input observation. It is equivalent to the local $K$-Lipschitz continuity condition over time space $t$, instead of input observation space $X$. $\delta$ time-varying condition is common for time-series tasks because they usually do not change abruptly.

**Online adaptation**   Online adaptation also can be called *adaptable prediction*, since it makes an inference concurring with updating model parameters [7, 9]. In online adaptation, the estimate of the model parameter is updated iteratively when new data is received. Online adaptation explores local overfitting to minimize the expected prediction error $e_{t+1}$, at time step $t$:

$$\mathcal{L}_{err}(t) = \min_\theta \mathbb{E}[e_{t+1}] = \min_\theta \mathbb{E}[\|Y_{t+1} - \hat{f}(\theta, X_t)\|_p], \tag{5}$$

where $Y_{t+1} = f(t, X_t) = [y_{t+1}, y_{t+2}, \cdots, y_{t+O}]$ is the ground truth observation (to be received in the future) and $\hat{Y}_{t+1} = \hat{f}(\theta_t, X_t)) = [\hat{y}_{t+1}, \hat{y}_{t+2}, \cdots, \hat{y}_{t+O}]$ is the predicted outcome from the learned model parameter $\theta_t$. The adaptation objective can be in any $\ell_p$ norm.

**Feedback adaptation**   As discussed above, the goal of the online adaptation is to minimize the expected prediction error in the future (5). Due to the lack of ground-truth value in the current steps, it is not feasible to directly minimize the prediction error. In **feedback adaptation**, the objective of minimizing the *prediction error* in the future is approximated by minimizing the *fitting error* in the past $\mathcal{L}_{fb}(t) = \min_{\theta_t} \|Y_t - \hat{f}(\theta_t, X_{t-1})\| + \lambda \mathcal{L}_{fb}(t-1) = \min_{\theta_t} \sum_{i=1}^{t} \lambda^{t-i} \|Y_i - \hat{f}(\theta_t, X_{i-1})\|$, where $0 \leq \lambda \leq 1$ is a forgetting factor. Then the optimization is done recursively. At time step $t$, after receiving the current observations $(x_t, y_t)$, the current samples were used to adjust the parameters of the prediction model by an online optimizer (e.g. SGD). Then a new prediction is made using the new optimized parameters. In the next time step, the estimate will be updated again given the new observation and the process repeats. In our main experiment and analysis, we utilize Stochastic Gradient Descent (SGD) as the optimizer. This selection is equivalent to setting $\lambda = 0$ in $\mathcal{L}_{fb}(t)$, resulting in the objective of optimizing the model using the latest observation. Additionally, we present results for the case where $\lambda = 1$, utilizing the Extended Kalman Filter (EKF) as the optimizer, as detailed in Appendix F.1.

Several prior works have aimed to enhance the robustness of feedback adaptation, with examples such as Abuduweili et al. [5], which employs multi-epoch training strategies to reuse hard samples, resulting in improved overall performance. However, it's important to note that feedback adaptation, in its current form, cannot fundamentally address the problem of forgetting. This limitation arises from the recent-based discounting in the objective function $\mathcal{L}_{fb}(t)$. To tackle this challenge, we propose a Feedforward Adaptation strategy with different objective functions, enabling the recall of important or similar past experiences. The next section will delve into the methodology in detail.

## 3   Methodology

### 3.1   General Online Adaptation Framework

Feedback adaptation recursively utilizes the latest observations to adapt its models, which may result in forgetting important past information. In order to enable recalling, we generalize the data compensation strategy of feedback adaptation (i.e. utilizing the latest observation) to samples from recent $L$-steps historical observations. The general online adaptation framework is shown in Algorithm 1. At time step $t$, after receiving the current observations $(x_t, y_t)$, we sample the critical input-output pairs $(X_s, y_{s+1})$ from recent $L$-steps historical observations. The critical pair will then be used to adjust the parameters of the model by an online optimizer (e.g. SGD). As a special case, the critical pairs are composed by the latest observations $X_s = X_{t-1}$ in feedback adaptation.

---

**Algorithm 1** General Online Adaptation Framework (Adaptable Prediction)

---

**Require:** Initial predictor $f(\theta_0, :)$ with parameters $\theta_0$, Optimizer $\mathcal{O}(:, :, :)$; buffer size $L$
**Ensure:** Sequence of predictions $\{\hat{Y}_{t+1}\}_{t=1}^T$
 1: **for** $t = 1, 2, \cdots, T$ **do**
 2:     Receive the ground truth observations $x_t, y_t$
 3:     Find the critical input-output pairs for adaptation $(X_s, y_{s+1})$, with $t - L \le s < t$
 4:     Adaptation: $\theta_t = \mathcal{O}(\theta_{t-1}, \hat{y}_{s+1}, y_{s+1})$, where $\hat{Y}_{s+1} = [\hat{y}_{s+1}, \cdots, \hat{y}_{s+O}] = \hat{f}(\theta_{t-1}, X_s)$
 5:     Prediction: $\hat{Y}_{t+1} = [\hat{y}_{t+1}, \cdots, \hat{y}_{t+O}] = \hat{f}(\theta_t, X_t)$
 6: **end for**

---

## 3.2 Feedforward Adaptation

We propose feedforward adaptation algorithms based on the feedforward compensation, which shares similarities with feedforward control strategies [24]. The proposed **feedforward adaptation** directly minimizes prediction error (bound) by compensating for historical data without relying on feedback mechanisms. The feedforward adaptation strategy works by estimating the error bound of the prediction model and selecting historical data that can compensate for the model to minimize this error bound. Specifically, when a prediction model encounters a sample resembling very early data, the feedforward adaptation method retrieves historically similar samples to enhance the learning process. By incorporating this mechanism, the model benefits from an improved ability to make predictions without disregarding past information, thereby achieving a minimum error bound.

Feedforward adaptation aims to minimize the *upper bound of the prediction error*:

$$\mathcal{L}_{ff}(t) = \min_{\theta_t} \mathrm{Bound}[e_{t+1}] = \min_{\theta_t} \mathrm{Bound}[\|Y_{t+1} - \hat{f}(\theta_t, X_t)\|]. \tag{6}$$

That is equivalent to optimizing the worst-case scenarios [25]. Minimizing the error bound of the prediction model is critical to providing robust prediction and safe planning [26]. To establish a sample selection strategy in feedforward compensation for minimizing the error bound, we estimate the error bound in Lemma 1.

**Lemma 1.** *Error Bound of Online Adaptation*. *Under the $K$-Lipschitz continuity condition and $\delta$ time-varying condition* (2) *to* (4)*, the (prior) prediction error $e_{t+1}$ of general online adaptation (Algorithm 1) has the following upper bound:*

$$e_{t+1} \le K\|X_t - X_s\| + \delta|t - s| + \|Y_{s+1} - \hat{f}(\theta_t, X_s)\| + \|\hat{f}(\theta_t, X_s) - \hat{f}(\theta_t, X_t)\| \tag{7}$$

$$\le (K + \hat{K})\|X_t - X_s\| + \delta|t - s| + \|Y_{s+1} - \hat{f}(\theta_t, X_s)\|. \tag{8}$$

The error bound estimation comprises three components. The first term accounts for the difference between the current sample $X_t$ and the selected critical sample $X_s$; the second term considers the time difference between current step $t$ and the step $s$ associated with the selected critical samples; the third term represents the (posterior) fitting error of input-output tuple $(X_s, Y_{s+1})$, which tends to be small through appropriate fitting techniques. The proof can be found in Appendix A.

Directly minimizing the error bound (8) is impractical since we do not exactly know the Lipschitz constant $K$ and time-varying factor $\delta$. However, in a slow time-varying system, it is possible to simplify the aforementioned error bound.

**Slow time-varying system** is one whose transition function (or behavior) changes slowly over time, i.e. $\delta \approx 0$. The slow time-varying system only assumes that the transition function $f$ varys slowly (locally) within recent $L$-steps, not (globally) for every step. Slow time-varying systems are common in real-world applications because we can choose the adaptation step to be small or the adaptation rate to be fast compared to the changing rate of the dynamics.[2] We describe the detailed condition

---

[2]In some cases, we can convert the non-slow-varying system into a slow-varying system by differencing, more details are shown in Section 4.2.

for feedforward adaptation in Section 4.1. In the slow time-varying system, let $\delta \approx 0$ in the error bound (8), the optimization objective for the error bound can be simplified as below:

$$\min_{\theta_t} \text{Bound}[e_{t+1}] = \min_{\theta_t, s}(K + \hat{K})\|X_t - X_s\| + \|Y_{s+1} - \hat{f}(\theta_t, X_s)\|, \tag{9}$$

The unknown parameter $K$ still makes it difficult to directly optimize the above objective. To address the problem, we change the joint minimization over $s$ and $\theta_t$ to a bi-level optimization which first minimizes the first term of objective (9) over the sampling time step $s$, then minimizes the second term of the objective over parameter $\theta_t$. Thus, the simplified objective function becomes:

$$\mathcal{L}_{final} = \min_{\theta_t} \|Y_{s^\star+1} - \hat{f}(\theta_t, X_{s^\star})\| \tag{10}$$

$$\text{s.t.} \quad s^\star = \arg\min_{i \in [t-L, t-1]} \|X_t - X_i\| \tag{11}$$

In summary, the proposed feedforward adaptation method operates as follows: it selects the most similar samples to the current observation to create the critical pair $(X_{s^\star}, Y_{s^\star+1})$, as determined by (11). Subsequently, it employs this critical pair to optimize the prediction model following the principles outlined in (10). When connected with Algorithm 1, this approach is essentially a replacement for the operations in line 4 and line 5 within Algorithm 1, substituting (11) and (10) in their place. For a comprehensive view of the feedforward adaptation algorithm, please refer to Appendix C.4.

### 3.3 Uncertainty Estimation

The error bound (7) provides uncertainty estimation of the prediction results. Here we use estimation of $\tilde{K}_t$ and $\tilde{\delta}$ to approximate real $K$ and $\delta$ in (7). We use confidence factor $\sigma \in (0, 1]$ to decay the error bound. The uncertainty estimation $\hat{U}_{t+1}$ for prediction $\hat{Y}_{t+1}$ can be computed as:

$$\hat{U}_{t+1} = \sigma \cdot (\tilde{K}_t\|X_t - X_s\| + \tilde{\delta}(t - s) + \|Y_{s+1} - \hat{f}(\theta_t, X_s)\| + \|\hat{f}(\theta_t, X_s) - \hat{f}(\theta_t, X_t)\|). \tag{12}$$

The confidence factor $\sigma$ is a predefined hyperparameter, e.g. $\sigma = 0.9$ for 90% confidence of uncertainty estimation. $\tilde{\delta}$ and $\tilde{K}_t$ are also predefined hyperparameters, in our experiments we set $\tilde{\delta} = 10^{-4}, \tilde{K} = 1$. In order to improve the accuracy of the uncertainty estimation, we have the option to iteratively update $\tilde{K}_t$ based on the estimated uncertainty $\hat{U}_t$ and the actual error $e_t$, such that if the previous uncertainty estimation is much larger than the real error then we shrink the $\tilde{K}_t$ value; and vice versa. The $\tilde{K}_t$ estimation rule are chosen to be:

$$\tilde{K}_t = \tilde{K}_{t-1} + \beta \cdot \frac{e_t - \hat{U}_t}{\|X_t - X_s\|}, \quad \beta \in [0, 1]. \tag{13}$$

## 4 Analysis

### 4.1 Comparison between Feedforward and Feedback Adaptation

The main difference between feedforward adaptation and feedback adaptation lies in the critical sample selection strategy that used in the optimization. In feedback adaptation, the critical sample is composed by the latest observation $X_s = X_{t-1}$. In the proposed feedforward adaptation, the critical sample is the most similar sample to the current observation within recent $L$-steps $X_s = \arg\min_{X_i}\|X_t - X_i\|$. In the following analysis, we assume there is no fitting error and mainly focus on the data selection strategy. The important findings regarding the expected error bounds for feedforward and feedback adaptations are presented in Lemma 2. Proof of these results can be found in Appendix B.

**Lemma 2.** *Expected Error Bound. Considering a transition function $f(t, X)$ and a parameterized model $\hat{f}(\theta, X)$ subject to the $K$-Lipschitz continuity condition and $\delta$ time-varying condition (2) to (4). Let $\mathbb{E}[\|X_t - X_{t-1}\|] := D$ represent the expected distance between consecutive samples, and $\mathbb{E}[\min_{X_i}|X_t - X_i|] := D^\star$ denote the expected minimum distance. We have the following results for error bound for feedback adaptation $B_e^{fb}$ and error bound for feedforward adaptation $B_e^{ff}$.*

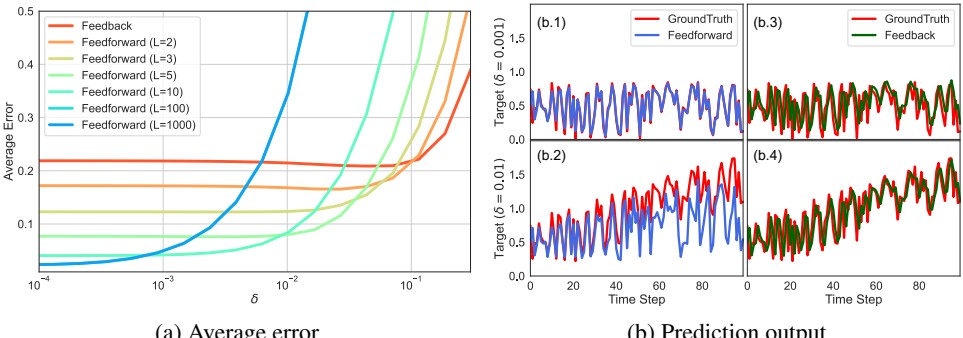

(a) Average error                                              (b) Prediction output

Figure 1: (a) Average error of feedback and feedforward adaptation with different buffer sizes $L$ and time-varying factors $\delta$. (b) Prediction output of feedforward (blue curve) and feedback adaptation (green curve) with buffer size $L = 100$ and time-varying factor $\delta = 10^{-3}$ and $\delta = 10^{-2}$.

(a) *Expected error bound for feedback adaptation:* $\mathbb{E}[B_e^{fb}] = (K + \hat{K})D + \delta$.

(b) *Expected error bound for feedforward adaptation:* $\mathbb{E}[B_e^{ff}] \leq (K + \hat{K})D^\star + \delta L$.

(c) *If time-varying factor $\delta$ is smaller, specifically $\frac{\delta}{K+\hat{K}} < \frac{D-D^\star}{L-1}$, feedforward adaptation has a smaller expected error bound than feedback adaptation.*

(d) *If the inputs are random variables sampled from the uniform distribution, $X_t \sim \mathcal{U}(0,1)$, we have more concrete conclusions. (1) $\mathbb{E}[B_e^{fb}] = \frac{K+\hat{K}}{3} + \delta$; (2) $\mathbb{E}[B_e^{ff}] = \frac{K+\hat{K}}{L+2} + \delta L$; (3) If $\frac{\delta}{K+\hat{K}} < \frac{1}{12}$, feedforward adaptation with buffer size $L = 2$ outperforms feedback adaptation. The optimal buffer size for minimum error bound is $L^\star = \sqrt{\frac{K+\hat{K}}{\delta}} - 2$.*

**Numerical Example**. We design a toy example to evaluate Lemma 2, where we use a neural network to learn the a linear time-varying system $y_{t+1} = f(x_t) = \sin x_t + \alpha t$ where $x_t \sim \mathcal{U}(0,1)$.

The results in Fig. 1a validate the theoretical results from Lemma 2(d). For a smaller time-varying factor $\delta$, feedforward adaptation with a larger buffer size achieves a smaller prediction error. However, if $\delta$ is larger, the performance of the feedforward adaptation deteriorates. If $\delta < 0.1$, feedforward adaptation (buffer size $L \geq 2$) outperforms feedback adaptation. Conversely, if $\delta > 0.1$, feedback adaptation is better. The threshold $\delta^\star = \frac{1}{12}(K + \hat{K}) \approx 0.1$ aligns with Lemma 2(d).

Figure 1b illustrate the comparison of prediction results between feedforward and feedback adaptation. The first row shows the prediction results for $\delta = 10^{-3}$ using feedforward adaptation with $L = 100$ (b.1) and feedback adaptation (b.3). As shown in the figure, feedforward adaptation results in better prediction. The second row shows the prediction results for $\delta = 10^{-2}$. The relatively poor performance observed in the feedforward approach depicted in Fig. 1b.2 aligns with our theoretical analysis. Compared to feedback adaptation, the feedforward adaptation method excels at learning the prediction model using critical samples, enabling it to effectively capture the input-dependent component, such as $\sin x_t$ in the numerical examples. However, it struggles to capture the time dependency, like $\alpha t$ in the numerical examples. Fortunately, the linear time dependency can be effectively mitigated through differencing (as discussed in the next section).

## 4.2 Applications of Feedforward Adaptation

According to Lemma 2(c), feedforward adaptation outperforms feedback adaptation when $\delta \approx 0$, regardless of the values of $\delta, K, D, D^\star$. To aid in making this determination, we propose a straightforward criterion based on time series stationarity analysis.

A stationary time series is one whose properties do not depend on the time at which the series is observed [27]. For a time series $(X_t, Y_{t+1})$, stationarity implies that the transition function $f : X_t \rightarrow Y_{t+1}$ does not have a significant time-varying factor $\delta$. To determine the stationarity of a given time series signal, we employ the ADF test [28]. If the time series is found to be stationary, indicating

$\delta \approx 0$, feedforward adaptation can be used to achieve improved results. If the original time series is non-stationary, we can employ differencing to transform the series and assess stationarity again based on the differenced signal. Differencing calculates the difference between two consecutive observations [29], i.e. predicting $d_{t+1} = Y_{t+1} - Y_t$ instead of $Y_{t+1}$. It stabilizes the mean of a time series and thus reduces the trend. Differencing has the ability to convert many non-stationary series into stationary ones. Readers are referred to Appendix C for more explanations.

## 5    Experiments

**Dataset**. In this section, we conduct a comprehensive evaluation of the proposed feedforward adaptation on both robotics-related and real-world time-series benchmarks. The robotics-related scenarios include the following tasks. 1) *Human Motion Prediction in Human-Robot Collaboration*: For this task, we utilize the THOR dataset [30] to predict human motion and the Assembly dataset [31] to predict arm motion. 2) *Vehicle Trajectory Prediction in Autonomous Driving*: We employ the NGSIM dataset [32] to assess vehicle trajectory prediction performance. 3) *Robotic Arm Trajectory Prediction for Quality Control and Monitoring*: The task involving the prediction of robotic arm trajectories in pick-and-place scenarios. Moreover, we evaluate the feedforward adaptation on three well-established real-world time-series benchmarks: 1) *Electricity Transformer Temperature (ETT) dataset* [33]; 2) *Exchange-Rate dataset* [34]; 3) *Influenza-like Illness (ILI) dataset*.

**Experimental Design**. We employ a Multi-layer Perceptron (MLP) with direct multistep (DMS) prediction strategy [35] as our parameterized prediction model $\hat{f}(\theta, :)$. We first train prediction models on the train set. In the evaluation phase, we simulate real-world applications by incrementally receiving observations from the test set. At each time step, we perform online adaptation by optimizing the model using stochastic gradient descent (SGD) with selected previous observations. Subsequently, we obtain the prediction output from the updated model. The prediction results are evaluated using the root-mean-squared error (RMSE) metric. More details of the experimental design can be found in Appendix D.

**Baselines**. We compare the proposed feedforward adaptation method with five baseline strategies. 1) *w/o adapt* performs prediction without any adaptation. 2) *Feedback adaptation* utilizes the latest observations to adapt its models [5]. 3) *Experience Replay (ER)* introduces random sample replay from this buffer to enhance learning [18]. 4) *Average Gradient Episodic Memory (A-GEM)* aims to minimize loss on current data under the constraint of avoiding loss increase on replayed data [19]. 5) *Sequential Monte Carlo Dropout (SMCD)* employs a particle filter to sustain a distribution over dropout masks, thereby dynamically adapting the neural network to changing environment [36].

**Results.** We conducted experiments across 10 distinct random seeds and present the mean results alongside their corresponding standard deviations. Results of the robotics-related datasets are shown in Table 1. Additionally, results for the general time-series prediction benchmarks are presented in Table 2. Evidently, the feedforward adaptation method consistently surpasses other baseline approaches across all datasets. This shows the effectiveness of the similarity-based sample selection strategy in online adaptation.

Table 1: Performance (RMSE) comparison between the proposed feedforward adaptation method and other baselines on Robotic-related datasets. We use **boldface** and underline for the best and second-best results.

| Method\Dataset | THOR (m) | Assembly (cm) | NGSIM (m) | Robot arm (rad) |
|---|---|---|---|---|
| w/o adapt | 1.208 ± 0.005 | 1.324 ± 0.001 | 1.203 ± 0.011 | 0.257 ± 0.001 |
| ER | 0.914 ± 0.002 | 1.188 ± 0.001 | 0.985 ± 0.015 | 0.210 ± 0.001 |
| A-GEM | 0.873 ± 0.002 | 1.194 ± 0.001 | 0.935 ± 0.012 | 0.216 ± 0.001 |
| SMCD | 0.937 ± 0.006 | 1.201 ± 0.001 | 1.002 ± 0.015 | 0.210 ± 0.002 |
| Feedback | 0.891 ± 0.002 | 1.191 ± 0.001 | 0.963 ± 0.012 | 0.214 ± 0.001 |
| Feedforward | **0.839 ± 0.002** | **1.180 ± 0.001** | **0.901 ± 0.011** | **0.193 ± 0.001** |

Table 2: Performance (RMSE) comparison between the proposed feedforward adaptation method and other baselines on general time-series benchmarks.

| Method\Dataset | ETTh1 | Exchange | ILI |
|---|---|---|---|
| w/o adapt | 0.485 ± 0.011 | 0.783 ± 0.006 | 2.195 ± 0.009 |
| ER | 0.391 ± 0.013 | 0.601 ± 0.013 | 1.943 ± 0.016 |
| A-GEM | 0.373 ± 0.013 | 0.619 ± 0.013 | 1.906 ± 0.012 |
| SMCD | 0.413 ± 0.015 | 0.673 ± 0.008 | 2.051 ± 0.010 |
| Feedback | 0.383 ± 0.013 | 0.619 ± 0.012 | 1.953 ± 0.013 |
| Feedforward | **0.357 ± 0.012** | **0.589 ± 0.012** | **1.843 ± 0.011** |

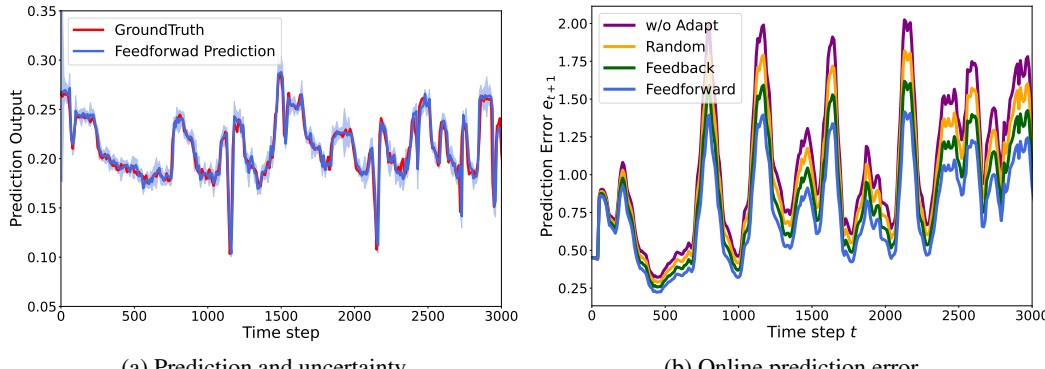

(a) Prediction and uncertainty                (b) Online prediction error

Figure 2: Experimental results on Assembly dataset. (a) A segment of the prediction output and uncertainty estimation of the proposed feedforward adaptation. For simplicity, only the prediction for the future 10th step in the first dimension is plotted. The blue dashed region represents the uncertainty. (b) Comparison of prediction errors between the feedforward adaptation and other baselines.

One advantage of the proposed feedforward adaptation is its capability to provide uncertainty estimation, as shown in (12) and (13). Figure 2a shows the prediction output (blue curve), ground truth label (red curve), and uncertainty estimation (blue dashed region) on the Assembly dataset. It is evident that the estimated uncertainty generally encompasses the actual ground truth value, confirming the effectiveness of the proposed uncertainty estimation. Figure 2b showcases the real prediction error for different adaptation methods over time, with feedforward adaptation exhibiting the smallest error. Moreover, in the context of periodic data, feedforward compensation inherently possesses the capacity to capture the underlying periodic patterns within the time series, as demonstrated in Appendix E.2. Furthermore, the proposed feedforward compensation strategy demonstrates the potential to generate additional improvements beyond the state-of-the-art optimization algorithms for online adaptation. This approach can be seamlessly integrated with any leading-edge prediction model, offering an avenue for further enhancements. More discussions are shown in Appendix F.

## 6    Conclusion, Limitation, and Future work

This paper investigates an effective feedforward adaptation algorithm for behavior prediction tasks. We provide evidence that feedforward adaptation exhibits a smaller error bound compared to conventional feedback adaptation in slow time-varying systems.

One of the limitations of the work is that the proposed feedforward adaptation method only works well on slow-varying systems. In the future, we plan to explore the integration of feedforward and feedback adaptation in more general systems, aiming for a more comprehensive approach and applications.

**Acknowledgments**

This research is supported by the National Science Foundation (NSF) under Grant No. 2144489.

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

## A    Lemma 1: Error Bound of Online Adaptation

In this section, we establish the error bound for prediction errors in the general online adaptation. At time step $t$, we select the critical input-output pairs $(X_s, y_{s+1})$ from recent $L$-steps observations.These critical pairs are utilized to update the parameters of the prediction model, resulting in a refined model. Subsequently, predictions are made using the newly optimized parameters.

Assuming that the transition function $f : X_t \to Y_{t+1}$ satisfies the $K$-Lipschitz continuity condition and the $\delta$ time-varying condition.

**Bound of ground-truth difference**. Given a transition function $f(t, X)$, if the $K$-Lipschitz continuity and $\delta$ time-varying conditions holds within recent $L$ steps, then the ground-truth value $Y_{t+1}$ and $Y_{s+1}$ has the following property:

$$\|Y_{t+1} - Y_{s+1}\| = \|f(t, X_t) - f(s, X_s)\| \le K\|X_t - X_s\| + \delta\|t - s\| \tag{14}$$

The proof is shown below:

$$\begin{aligned}
\|Y_{t+1} - Y_{s+1}\| &= \|f(t, X_t) - f(s, X_s)\| \\
&= \|f(t, X_t) - f(t, X_s) + f(t, X_s) - f(s, X_s)\| \\
&\le \|f(t, X_t) - f(t, X_s)\| + \|f(t, X_s) - f(s, X_s)\| \quad \text{(triangle inequality)} \\
&\le K\|X_t - X_s\| + \|f(t, X_s) - f(s, X_s)\| \quad \text{(K Lipschitzness)} \\
&\le K\|X_t - X_s\| + \delta|t - s| \quad \text{(}\delta\text{ time varying)}
\end{aligned} \tag{15}$$

**Error Bound of Online Adaptation**.  For time step $t$, the (prior) prediction error $e_{t+1}$ has the following inequality:

$$\begin{aligned}
e_{t+1} &= \|Y_{t+1} - \hat{Y}_{t+1}\| = \|Y_{t+1} - \hat{f}(\theta_t, X_t)\| \\
&= \|Y_{t+1} - Y_{s+1} + Y_{s+1} - \hat{f}(\theta_t, X_s) + \hat{f}(\theta_t, X_s) - \hat{f}(\theta_t, X_t)\| \\
&\le \|Y_{t+1} - Y_{s+1}\| + \|Y_{s+1} - \hat{f}(\theta_t, X_s)\| + \|\hat{f}(\theta_t, X_s) - \hat{f}(\theta_t, X_t)\| \quad \text{(triangle inequality)} \\
&\le K\|X_t - X_s\| + \delta|t - s| + \|Y_{s+1} - \hat{f}(\theta_t, X_s)\| + \|\hat{f}(\theta_t, X_s) - \hat{f}(\theta_t, X_t)\|
\end{aligned} \tag{16}$$

The first two terms come from the difference between ground-truth $Y_{t+1} - Y_{s+1}$, the third term is a (posterior) fitting error for input-output tuple $(X_s, Y_{s+1})$, and the last term is the difference between two predictions. Combining the above inequality with the Lipschitz continuity condition for $\hat{f}(\theta_t, X_t)$, we obtain the error bound for general online adaptation is shown below:

$$e_{t+1} \le (K + \hat{K})\|X_t - X_s\| + \delta|t - s| + \|Y_{s+1} - \hat{f}(\theta_t, X_s)\| \tag{17}$$

Then lemma 1 is derived.

## B    Comparison of Error Bound between Feedforward Adaptation and Feedback Adaptation

### B.1    Lemma 2 (a,b,c): Expected Error Bound

Considering the error bound (17), the first two terms are associated with the specific data compensation strategy, while the last term represents the posterior fitting error on the selected samples. In this study, our main focus is on the data compensation strategy, and we do not prioritize the data fitting aspect. Additionally, with a powerful neural network prediction model, achieving a very small fitting error (almost zero) is relatively straightforward [37]. Therefore, we can disregard the fitting error when comparing feedforward and feedback adaptation. By neglecting the fitting error, we obtain an approximate upper bound $B_e$ for general online adaptation, as shown below:

$$B_e = (K + \hat{K})\|X_t - X_s\| + \delta|t - s| \tag{18}$$

**Error Bound for Feedforward Adaptation**. In feedforward adaptation, the selected input-output pairs are the most similar samples to the current observation $X_s = \arg\min_{X_i} \|X_t - X_i\|$ from

$L$-size buffer, and $s = \arg\min_{i \in [t-L, t-1]} \|X_t - X_i\|$. Then we have an error bound $B_e^{ff}$ for feedforward adaptation:

$$B_e^{ff} = (K + \hat{K})\|X_t - X_s\| + \delta|t - s| \le (K + \hat{K})\|X_t - X_s\| + \delta L \tag{19}$$

$$X_s = \arg\min_{X_i \in [X_{t-L}, X_{t-1}]} \|X_t - X_i\| \tag{20}$$

**Error Bound for Feedback Adaptation**. In feedback adaptation, the selected input-output pairs are the latest observations $X_s = X_{t-1}$ and $s = t-1$. Then we have an error bound $B_e^{fb}$ for feedforward adaptation:

$$B_e^{fb} = (K + \hat{K})\|X_t - X_{t-1}\| + \delta \tag{21}$$

**Comparison of the expected error bound between Feedforward and Feedback Adaptation**. Let the expected distance between consecutive samples is $D$:

$$D := \mathbb{E}[\|X_t - X_{t-1}\|]. \tag{22}$$

Let the expected minimum sample distance is $D^\star$:

$$D^\star := \mathbb{E}[\|X_t - X_s\|] = \mathbb{E}[\min_{X_i} \|X_t - X_i\|]. \tag{23}$$

Then the expected error bound for feedforward adaptation is:

$$\mathbb{E}[B_e^{ff}] \le (K + \hat{K})\mathbb{E}[\|X_t - X_s\|] + \delta L = (K + \hat{K})D^\star + \delta L. \tag{24}$$

The expected error bound for feedback adaptation is:

$$\mathbb{E}[B_e^{fb}] = (K + \hat{K})\mathbb{E}[\|X_t - X_{t-1}\|] + \delta = (K + \hat{K})D + \delta. \tag{25}$$

Consider the conditions that feedforward adaptation has a smaller error bound than feedback adaptation in expectation. In order to make: $\mathbb{E}[B_e^{ff}] < \mathbb{E}[B_e^{fb}]$, we have:

$$(K + \hat{K})D^\star + \delta L < (K + \hat{K})D + \delta \tag{26}$$

$$\Rightarrow \frac{\delta}{K + \hat{K}} < \frac{D - D^\star}{L - 1} \tag{27}$$

Equation (27) represents the condition under which feedforward adaptation surpasses feedback adaptation in terms of the expected error bound. Here, the hyperparameter $L$ denotes the predefined buffer size. It is important to note that when $L = 1$, feedforward adaptation is equivalent to feedback adaptation. Therefore, our focus is primarily on the case when $L > 1$. From the equation, we observe that if the system exhibits a smaller time-varying property $\delta$ compared to the Lipschitz constant $K$, and a smaller minimum sample distance $D^\star$, feedforward adaptation is more likely to achieve a greater improvement over feedback adaptation. For instance, when $\delta = 0$, we have $\mathbb{E}[B_e^{ff}] < \mathbb{E}[B_e^{fb}]$ for any $K, \hat{K}, D, D^\star$, and $L$.

By combining (24), (25) and (27), we can conclude Lemma 2 (a,b,c).

## B.2   Lemma 2 (d): Expected Error Bound on Random-input System

Consider a transition function $f$ with randomly sampled input observations. Specifically, input $X_t$ is a random variable sampled from the uniform distribution: $X_t \sim \mathcal{U}(0, 1)$. In this case, the current sample $X_t$ and last sample $X_{t-1}$ are independent random variables from $\mathcal{U}(0, 1)$. According to [38], the expectation of the distance between these two independent and uniform-distributed variables is $\frac{1}{3}$. Then for feedback adaptation

$$\mathbb{E}[\|X_t - X_{t-1}\|] = \frac{1}{3}, \text{ for } X_t, X_{t-1} \sim \mathcal{U}(0, 1) \tag{28}$$

The term $E[\min_{X_i} \|X_t - X_i\|]$ represents the expected minimum distance between the current sample $X_t$ and previous $L$ samples in the buffer, which is $\frac{1}{L+2}$ [38], according to [38]. Then for feedforward adaptation:

$$\mathbb{E}[\|X_t - X_s\|] = \mathbb{E}[\min_{X_i \in [X_{t-L}, X_{t-1}]} \|X_t - X_i\|] = \frac{1}{L+2}, \text{ for } X_t, X_i \sim \mathcal{U}(0, 1) \tag{29}$$

Let $D = \frac{1}{3}, D^\star = \frac{1}{L+2}$ on the expected error bound (24) and (25), we obtain the expected error bound for feedforward and feedback adaptation on the system with random input:

$$\mathbb{E}[B_e^{ff}] = (K + \hat{K})D^\star + \delta L = \frac{K + \hat{K}}{L + 2} + \delta L \tag{30}$$

$$\mathbb{E}[B_e^{fb}] = (K + \hat{K})D + \delta = \frac{K + \hat{K}}{3} + \delta \tag{31}$$

Consider the conditions that feedforward adaptation has a smaller error bound than feedback adaptation in expectation. In order to make: $\mathbb{E}[B_e^{ff}] < \mathbb{E}[B_e^{fb}]$, we have:

$$(K + \hat{K})D^\star + \delta L < (K + \hat{K})D + \delta \tag{32}$$

$$\Rightarrow \frac{K + \hat{K}}{L + 2} + \delta L < \frac{K + \hat{K}}{3} + \delta \tag{33}$$

$$\Rightarrow \frac{\delta}{K + \hat{K}} < \frac{1}{3L + 6} \tag{34}$$

If $L = 1$, the feedforward adaptation is equal to the feedback adaptation. For feedforward adaptation, we have $L > 1$. Then we consider the buffer size $L = 2$ as general settings, then conclude the conditions for applying feedforward adaptation:

$$\frac{\delta}{K + \hat{K}} < \frac{1}{3L + 6} = \frac{1}{12} \approx 0.083 \tag{35}$$

In this case, with the optimal buffer size $L = L^\star := \sqrt{\frac{K+\hat{K}}{\delta}} - 2$, feedforward adaptation achieves the smallest expected error bound:

$$\mathbb{E}[B_e^{ff}]^\star = 2\sqrt{\delta(K + \hat{K})} - 2\delta \tag{36}$$

As can be seen, if $\delta \approx 0$, feedforward adaptation could achieve the zero expected error bound with optimal buffer size $L^\star$, while feedback adaptation cannot converge to zero expected error bound.

Thus, given a prediction system $f$ with a random input state, if $\frac{\delta}{K+\hat{K}} < \frac{1}{12}$, with buffer size $L = 2$, feedforward adaptation achieves the smaller expected error bound than feedback adaptation. In this case, the optimal buffer size for minimum error bound is $L^\star = \sqrt{\frac{K+\hat{K}}{\delta}} - 2$.

By combining (30), (31), (35) and (36), one can conclude Lemma 2 (d).

## B.3 Synthetic Experiments: Linear Time-varying System

We design a toy experiment to evaluate Lemma 2. We consider the following linear time-varying system

$$y_{t+1} = f(x_t) = \sin x_t + \delta t, \quad x_t \sim \mathcal{U}(0, 1)$$

Our parameterized prediction model is a one-layer perception with Sigmoid activation function.

$$\hat{y}_t = \hat{f}(V_t, b_t; x_t) = S(V_t x_t) + b_t = \frac{1}{1 + e^{-V_t x_t}} + b_t \tag{37}$$

Where $S(\cdot)$ denotes a Sigmoid activation function. We have The Lipschitz constant $K$ and $\hat{K}$ for the ground-truth function $f$ and the one-layer perception $\hat{f}$:

$$K = \sup |\frac{\partial f}{\partial x_t}| = \sup |\cos(x_t)| = 1 \tag{38}$$

$$\hat{K} = \sup(|\frac{\partial \hat{f}}{\partial x_t}|) = \sup |V_t \cdot S(V_t x_t) \cdot (1 - S(V_t x_t))| = 0.25 \sup |V_t| \tag{39}$$

We use SGD as an optimizer in feedback and feedforward adaptation. During training, we keep the $\|V_t\|$ bounded, i.e. $\|V_t\| \leq 1$, then $\hat{K} = 0.25$. We use Lemma 3 (30) and (31) to calculate the error bound for feedback and feedforward adaptation:

$$\mathbb{E}[B_e^{fb}] = \frac{5}{12} + \delta \tag{40}$$

$$\mathbb{E}[B_e^{ff}] = \frac{5}{4L + 8} + \delta L \tag{41}$$

Then we calculate the threshold $\delta^\star$ (35). If $\delta \leq \delta^\star$, feedforward adaptation has a smaller error bound.

$$\frac{\delta^\star}{K + \hat{K}} = \frac{1}{12} \tag{42}$$

$$\Rightarrow \delta^\star = \frac{1}{12}(K + \hat{K}) \approx 0.1 \tag{43}$$

Thus, in the toy experiment, If $\delta \leq 0.1$, feedforward adaptation has a smaller error bound. The experimental results are shown in Figure 1 of the main paper.

## C  Applications of Feedforward Adaptation

When determining whether to apply feedforward adaptation to a system or time-series function, Lemma 2(c) can serve as a criterion. However, estimating the values of $\delta, K, D, D^\star$ for the system is required. As a straightforward and conservative approach, if $\delta \approx 0$, feedforward adaptation outperforms feedback adaptation for any $\delta, K, D, D^\star$. To simplify this decision-making process, we propose a simple criterion based on the widely used stationarity test in time-series analysis.

### C.1  Stationary time series and ADF test

A stationary time series is one that exhibits properties that do not depend on time. Therefore, a stationary time series does not possess trends or seasonality. In the context of a time series $(X_t, Y_{t+1})$, stationarity implies that the transition function $f : X_t \to Y_{t+1}$ is not explicitly linked to the time step $t$. In accordance with the $\delta$ time-varying condition, which is equivalent to $\delta \approx 0$.

The Augmented Dickey-Fuller (ADF) test is a widely used method for detecting the stationarity of a time series [28]. It tests the null hypothesis that a time series is non-stationary or time-dependent (i.e., it has a unit root), while the alternative hypothesis suggests stationarity, indicating that it cannot be represented by a unit root. The ADF test yields a p-value that is used to assess the test. As the proposed feedforward algorithm does not necessitate an exceedingly strict stationary condition, we have set the threshold between stationary and non-stationary in our task to be 0.1. If the p-value is less than 0.1, we reject the null hypothesis and conclude that the series is stationary. Conversely, if the p-value is greater than or equal to 0.1, we fail to reject the null hypothesis and conclude that the series is non-stationary.

### C.2  Differencing

In many real-world scenarios, time series signals exhibit non-stationarity. Therefore, it is crucial to transform these non-stationary signals into stationary ones in order to apply feedforward adaptation effectively. One approach to achieve this is by computing the differences between consecutive observations, denoted as $d_{t+1} = Y_{t+1} - Y_t$. This process is commonly referred to as differencing [29]. Differencing helps stabilize the mean of a time series by eliminating changes in its level and removing trends. By applying differencing, it becomes possible to convert many non-stationary series into stationary ones, thereby facilitating the use of feedforward adaptation.

### C.3  Criterion for applying feedforward adaptation

The criterion and procedure for applying feedforward adaptation are presented in Figure 2 of the main paper. In this approach, given a time series $(X_t, Y_{t+1})$, such as the training set of the prediction

task $f : X_t \rightarrow Y_{t+1}$, we follow a specific process based on the stationarity of the series. If the ADF test indicates that the series is stationary, we directly apply feedforward adaptation to the original series. This involves prediction and adaptation on $Y_{t+1} = f(t, X_t)$. If the series is found to be non-stationary, we employ differencing by calculating the difference between consecutive observations, denoted as $d_{t+1} = Y_{t+1} - Y_t$. We then assess the stationarity of the differenced signal $d_{t+1}$. If it is determined to be stationary, we proceed with feedforward adaptation on the difference series. This entails prediction and adaptation on $d_{t+1}$, followed by converting it back to $Y_{t+1} = Y_t + d_{t+1}$ based on the value of $Y_t$. If the differenced signal remains non-stationary even after differencing, we resort to feedback adaptation for handling the non-stationary signal. The criterion and procedure for applying feedforward adaptation are shown in Fig. 3. Please note that the differencing operator can be applied up to $K \geq 1$ times. In our experiments, we utilize at most one differencing operation, which is equivalent to predicting velocity in trajectory datasets.

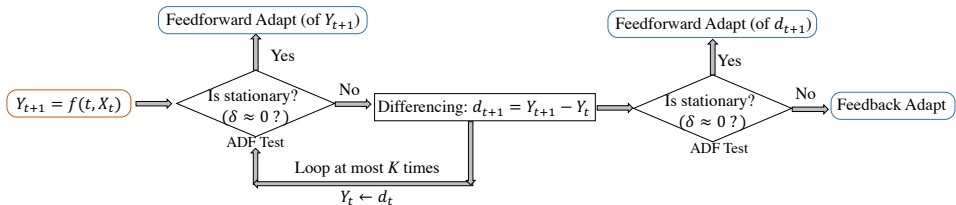

Figure 3: The criterion and procedure for applying feedforward adaptation.

## C.4 Feedforward Adaptation Algorithms

---
**Algorithm 2** Online Adaptation with Feedforward Compensation

---
**Require:** Initial predictor $f(\theta_0, :)$ with parameters $\theta_0$, Optimizer $\mathcal{O}(:, :, :)$, $L$-size buffer $B$
**Ensure:** Sequence of predictions $\{\hat{Y}_{t+1}\}_{t=1}^T$ and estimated uncertainty $\{\hat{U}_{t+1}\}_{t=1}^T$
1: **for** $t = 1, 2, \cdots, T$ **do**
2:     Receive the ground truth observation values $x_t, y_t$; Construct input $X_t = [x_{t-I+1}, \cdots, x_t]$
3:     Find the critical (similar) input-output pairs $(X_{s^\star}, y_{s^\star+1})$ from buffer $B$ by (11)
4:     Adaptation by (10): $\theta_t = \mathcal{O}(\theta_{t-1}, \hat{y}_{s^\star+1}, y_{s^\star+1})$
5:     Prediction: $\hat{Y}_{t+1} = [\hat{y}_{t+1}, \cdots, \hat{y}_{t+O}] = f(\theta_t, X_t)$
6:     Uncertainty $\hat{U}_{t+1}$ Estimation by (12) and (13)
7:     Add current data to buffer: $B.append(X_t, y_t)$
8:     **if** $size(B) > L$ **then**
9:         $B \leftarrow$ keep_more_recent_samples $(B, L)$
10:     **end if**
11: **end for**

---

# D  Additional Details of Experimental Design

## D.1  Dataset

We evaluate the effectiveness of the proposed feedforward adaptation method in three robotic-related scenarios: (1) Human motion prediction in human-robot collaboration, using the THOR dataset and Assembly dataset; (2) Vehicle trajectory prediction in autonomous driving, using the NGSIM dataset; and (3) Robotic arm trajectory prediction for quality control and monitoring purposes, using the Robot arm trajectory dataset. The specific tasks for each dataset are illustrated in Fig. 4.

The description of the robotic-related datasets is shown below.

- **THOR**[3] is a public dataset of human motion trajectories, recorded in a controlled indoor experiment [30]. Which includes the motion trajectories with diverse and accurate social human motion

---
[3]http://thor.oru.se/

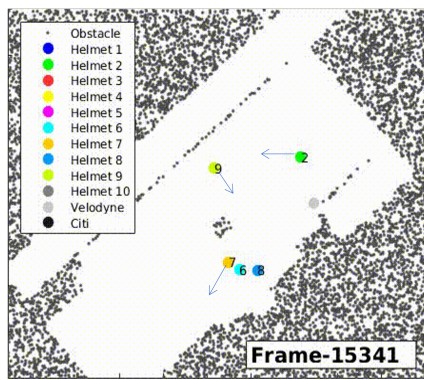

(a) THOR human motion prediction dataset

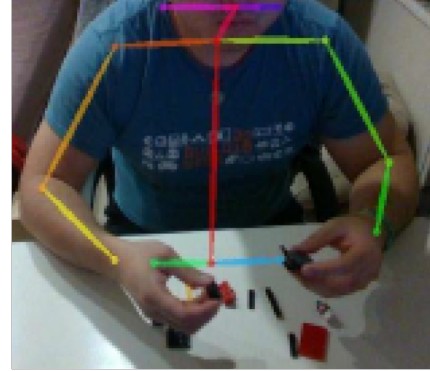

(b) Arm motion prediction in assembly tasks

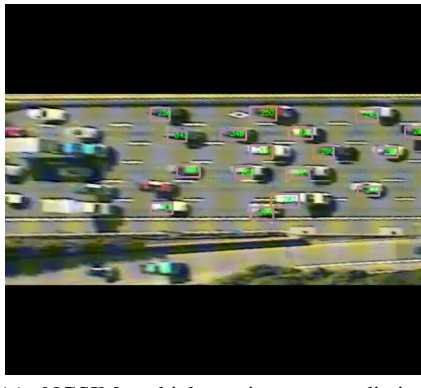

(c) NGSIM vehicle trajectory prediction dataset

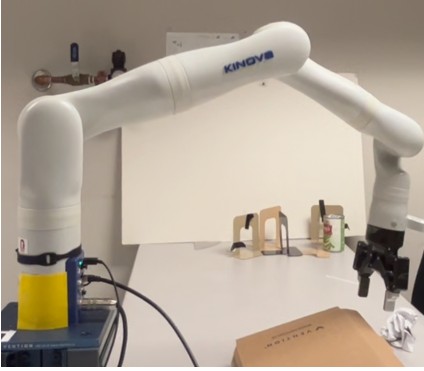

(d) Robot arm trajectory prediction in pick-and-place tasks

Figure 4: Illustration of tasks in different datasets. Figure (a) is copied from the public website of the THOR dataset `http://thor.oru.se/`; Figure (b) is copied from the website of the Assembly dataset `https://github.com/intelligent-control-lab/Human_Assembly_Data`; Figure (c) is copied from the public website of the NGSIM dataset `https://data.transportation.gov/Automobiles/Next-Generation-Simulation-NGSIM-Program-I-80-Vide/2577-gpny`; Figure (d) shows a KINOVA robot arm performing pick-and-place tasks in our dataset.

 

data in a shared indoor environment. In our experiments, we use No. $2 \sim 4$ agent's trajectory as a train set and No. $5 \sim 10$ agent's trajectory as a test set.

- **Assembly** dataset [4] records arm motions in assembly tasks. This dataset includes 5 different assembly tasks. Each task requires the human to use LEGO pieces to assemble an object. In our experiments, we use task $1 \sim 2$ as a train set and task $3 \sim 5$ as a test set.

- **NGSIM** dataset: US 101 human driving data from Next Generation SIMulation dataset [5]. The dataset contains highway driving trajectories captured by cameras mounted on top of surrounding buildings [32]. In our experiment, we use a subset of the dataset which contains 100 trials of different agents. We use No. $1 \sim 50$ trial's trajectory as a train set and No. $50 \sim 100$ trial's trajectory as a test set.

- We collect the **Robot arm trajectory** dataset, which records the joint position (Denavit–Hartenberg parameters) of the KINOVA Gen 3 (7 DoF) robotic arm in pick-and-place tasks. This dataset includes 4 pick-and-place tasks for picking objects from different positions on a workbench. In our experiments, we use task $1 \sim 2$ as a train set and task $2 \sim 4$ as a test set.

---

[4] `https://github.com/intelligent-control-lab/Human_Assembly_Data`
[5] `https://www.fhwa.dot.gov/publications/research/operations/07030/index.cfm`

In addition to the evaluation of four robotics-related datasets, we further assess the proposed feed-forward adaptation method using three real-world time-series benchmarks: ETT (Electricity Transformer Temperature), Exchange-Rate, and ILI (Influenza-like Illness) dataset. These three datasets are extensively employed within the domain of time-series prediction research. The description of the time-series benchmarks datasets is shown below.

- **ETT** [33] dataset contains the data collected from electricity transformers, including load and oil temperature that are recorded every 15 minutes between July 2016 and July 2018. Which consists of two hourly-level datasets (ETTh) and two 15-minute-level datasets (ETTm). In our experiments, we used the first hourly-level dataset ETTh1 as a univariance prediction task.

- **Exchange-Rate** [34] records the daily exchange rates of eight countries from 1990 to 2016.

- **ILI** [6] describes the ratio of patients seen with ILI and the total number of patients. Which includes the weekly recorded influenza-like illness (ILI) patients data from the Centers for Disease Control and Prevention of the United States between 2002 and 2021.

## D.2 Stationarity test of datasets

As discussed in Appendix C, we use the ADF method to test the stationarity of the time-series data and check the slow-varying property of its transition function. If the p-value of the ADF test is less than 0.1, we can reject the null hypothesis and conclude that the time series is stationary. If the p-value of the ADF test is greater than 0.1, we cannot reject the null hypothesis and conclude that the time series is non-stationary.

Table 3: ADF test results for raw time-series and the difference signal on Thor, Assembly, NGSIM, Robot arm datasets.

| Dataset | THOR | Assembly | NGSIM | Robot arm |
|---|---|---|---|---|
| P value on Raw Series | 0.005 (stationary) | 4e-3 (stationary) | 0.34 (nonstationary) | 0.09 (stationary) |
| P value on Difference | < 0.001 | < 0.001 | 0 (stationary) | 0.008 |

Table 4: ADF test results for ETTh1, Exchange-rate, and ILI datasets.

| Dataset | ETTh1 | Exchange-rate | ILI |
|---|---|---|---|
| P value on Raw Series | 0.008 (stationary) | 0.533 (non-stationary) | 0.049 (stationary) |
| P value on Difference | 0 | 0 | < 0.001 |

The outcomes of the Augmented Dickey-Fuller (ADF) test on the robotics-related datasets are presented in Table 3. It is evident that the original trajectories (raw signals) for the THOR, Assembly, and Robot Arm datasets exhibit stationarity. This suggests that the transition function for these datasets changes slowly over time. On the other hand, for the NGSIM dataset, the original trajectory is non-stationary, but the difference signal (velocity) is stationary. In our experiments, we apply feedforward adaptation to the raw trajectory for the THOR, Assembly, and Robot Arm datasets, and we apply feedforward adaptation to velocity (difference signal) for the NGSIM dataset.

In the case of general time-series datasets, to maintain consistency with previous works [33] that directly predict raw time-series signals (without differencing), we employ feedforward adaptation on the original raw signals. For reference, the ADF test results of the general time-series datasets are depicted in Table 4.

## D.3 Experimental design

**Parameterized Prediction models**. We utilize a Multi-layer Perceptron (MLP) with a direct multistep (DMS) prediction strategy [35]. The choice of MLP with DMS is motivated by the superior performance of a simple MLP over many larger Transformer-based models, as reported in [35]. Our

---

[6]https://gis.cdc.gov/grasp/fluview/fluportaldashboard.html

MLP architecture consists of two layers. The first layer can be considered as an Encoder, denoted as $X_t = W \cdot X_t$. Following the encoder, the MLP incorporates layer normalization, an activation function, and a final linear projection represented as $Y_{t+1} = V \cdot \text{Relu}(\text{LayerNorm}(X_t))$. The layer normalization and the final projection can be viewed as a decoder. It is worth noting that we do not flatten the input for the MLP. The expression $X_t = W \cdot X_t$ represents a linear layer applied along the temporal axis.

**Baselines**. We compare the proposed feedforward adaptation method with five baseline strategies.

- *w/o adapt* performs prediction without any adaptation, serving as a lower bound for all adaptation methods.

- *Feedback adaptation* is a widely used online adaptation strategy, which utilizes the latest observations to adapt its models and minimize the fitting error for the last observations [5].

- *Experience Replay (ER)* is a widely used replay-based continual learning approach [18]. ER maintains a compact memory buffer containing select old training samples. During each iteration, ER introduces random sample replay from this buffer to enhance learning.

- *Average Gradient Episodic Memory (A-GEM)* integrates memory-relay and constrained optimization [19]. Similar to ER, A-GEM employs random data replay from the memory buffer. However, A-GEM's unique aspect is that its objective doesn't directly minimize loss on replayed samples. Instead, it aims to minimize loss on current data under the constraint of avoiding loss increase on replayed data.

- *Sequential Monte Carlo Dropout (SMCD)* is a simple and effective approach to adapting neural models in response to changing settings [36]. SMCD treats learning a network with a dropout layer akin to learning an ensemble of prediction distribution. At the adaptation phase, SMCD employs a particle filter to sustain a distribution over dropout masks, thereby dynamically adapting the neural model to evolving settings.

**Hyperparameters**. For offline training, we follow the strategy in [35]. In adaptation, we set the learning rate of SGD as $\eta = 0.001$. Buffer size for feedforward adaptation is $L = 1000$. For uncertainty estimation, we set $\tilde{\delta} = 0, \tilde{K} = 1$.

**Input and output horizon of prediction task.** For the robotics-related datasets, the prediction model utilizes the most recent 1 second of observations to predict the trajectory for the next 2 seconds. To ensure consistent sampling frequencies, we subsampled the THOR and Assembly datasets to 20Hz. For these datasets, we set the input horizon to 20 and the prediction horizon to 40. The NGSIM dataset has a sampling frequency of 15Hz, so we adjusted the input horizon to 15 and the prediction horizon to 30 accordingly. As for the Robot arm trajectory dataset, we subsampled it to a sampling frequency of 25Hz and set the input horizon to 25 and the prediction horizon to 50.

**Adaptation and Evaluation**. We first train prediction models on the train set. In the evaluation phase, we simulate real-world applications by incrementally receiving observations from the test set. At each time step, online adaptation is used to optimize the normalization layers of the model using stochastic gradient descent (SGD) with selected previous observations. To leverage the pretrained feature extraction part of the model without changing the parameters of the entire model, we only update the parameters of the normalization layer. Subsequently, we obtain the prediction output from the updated model. The prediction results are evaluated using the root-mean-squared error (RMSE) metric.

Concerning the real-world time-series benchmarks, we align our settings with those presented in the LTSF-benchmark [35]. The ETTh1 and Exchange-rate datasets share an input horizon of $I = 96$ and an output horizon of $O = 192$. Conversely, the ILI dataset entails an input horizon of $I = 36$ and an output horizon of $O = 36$.

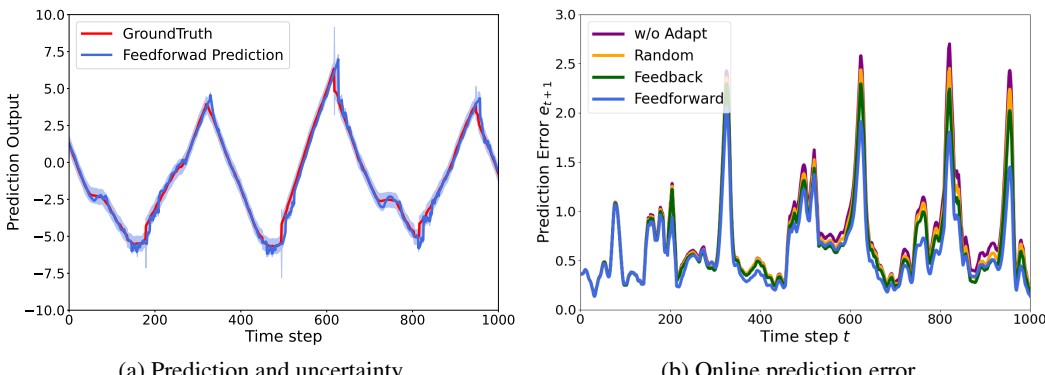

(a) Prediction and uncertainty        (b) Online prediction error

Figure 5: Experimental results on THOR dataset.

## E    Additional Experimental Results

### E.1    Prediction output and Prediction Error

Figure 5a shows the prediction output (blue curve), ground truth label (red curve), and uncertainty estimation (blue dashed region) on the THOR dataset. Figure 5b shows the real prediction error for different adaptation methods over time. Notably, feedforward adaptation exhibits the lowest prediction error among them.

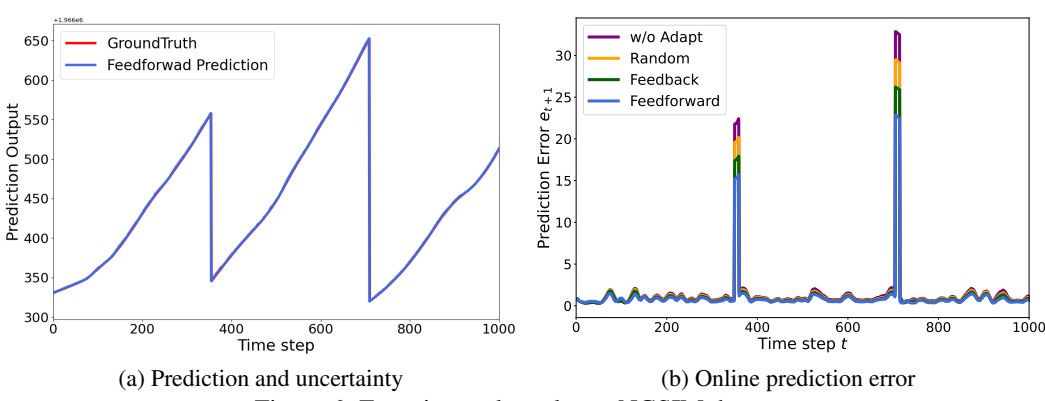

(a) Prediction and uncertainty        (b) Online prediction error

Figure 6: Experimental results on NGSIM dataset.

Figure 6a shows the prediction output (blue curve), ground truth label (red curve), and uncertainty estimation (blue dashed region) on the NGSIM dataset. Figure 6b shows the real prediction error for different adaptation methods over time. Notably, feedforward adaptation exhibits the lowest prediction error among them.

Figure 7a shows the prediction output (blue curve), ground truth label (red curve), and uncertainty estimation (blue dashed region) on the Robot arm dataset. Figure 7b shows the real prediction error for different adaptation methods over time. Notably, feedforward adaptation exhibits the lowest prediction error among them.

Figure 8a shows the prediction output (blue curve), ground truth label (red curve), and uncertainty estimation (blue dashed region) on the Etth1 dataset. Figure 8b shows the real prediction error for different adaptation methods over time. Notably, feedforward adaptation exhibits the lowest prediction error among them.

### E.2    Study of the sample selection strategy of different adaptation methods

Feedforward adaptation selects samples with the smallest sample difference $\min_{X_i} |X_t - X_i|$. This selection strategy allows feedforward adaptation to inherently capture the periodicity in time-series

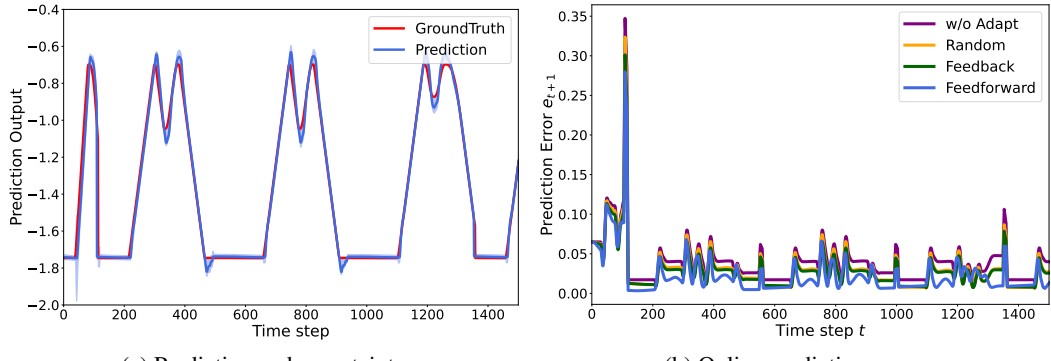

(a) Prediction and uncertainty

(b) Online prediction error

Figure 7: Experimental results on Robot arm dataset.

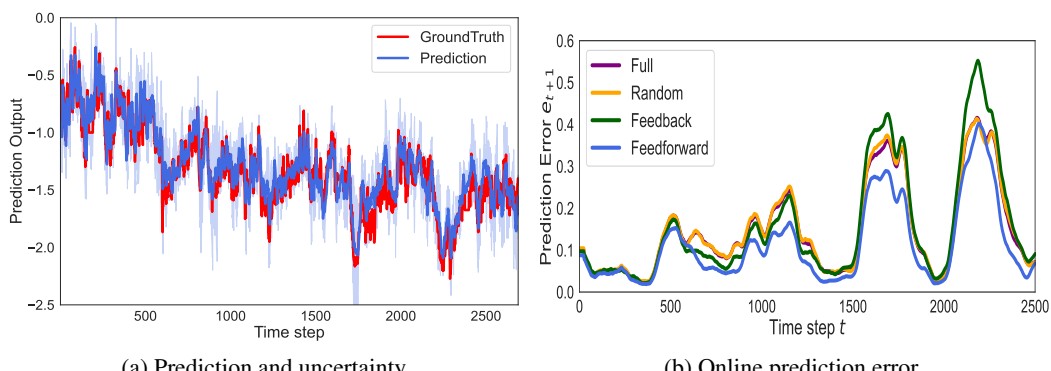

(a) Prediction and uncertainty

(b) Online prediction error

Figure 8: Experimental results on ETTh1 dataset.

data when faced with periodic patterns. In the case of the robot arm dataset, as depicted in Figure 7a, we observe an approximate periodicity of $T \approx 420$. This is evident from the FFT (Fast Fourier Transform) period analysis depicted in Figure 9a. In Figure 9b, we demonstrate how many samples were chosen from $(t-s) \approx 420$ steps earlier during the feedforward compensation process, aligning with the repetition period of $T \approx 420$. Feedforward adaptation's selection of the most similar samples to the current sample facilitates the extraction of hidden periodic patterns within the input signal over time. Consequently, the distribution of $(t - s)$ exhibits similarity to the FFT period analysis.

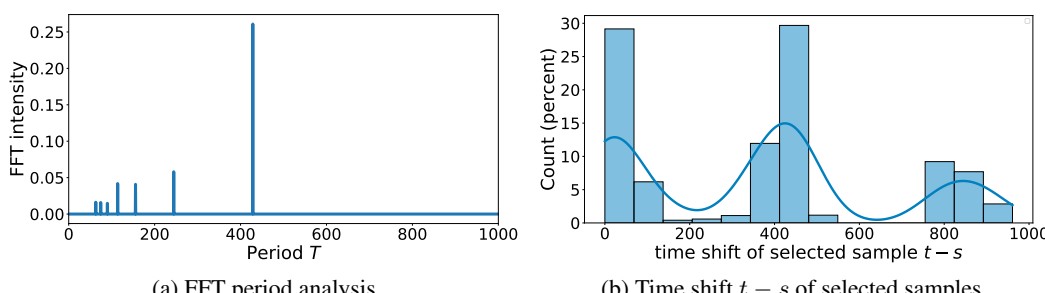

(a) FFT period analysis

(b) Time shift $t - s$ of selected samples

Figure 9: Experimental results on Robot arm dataset. (a) FFT period analysis. (b) Timeshift $t - s$ between current sample $X_t$ and selected sample $X_s$ in feedforward adaptation.

# F Discussion

## F.1 Incorporating Feedforward Adaptation with SOTA optimization algorithms

The proposed feedforward compensation strategy exhibits the capability to yield further enhancements on top of the SOTA optimization algorithms for online adaptation. Many prior works consider the information gain in a feedback fashion. Here we specifically compare with MEKF [14]. The MEKF introduces an Extended Kalman Filter (EKF)-based methodology for effectively leveraging samples to achieve reduced fitting errors. Nevertheless, this paper [14] still incorporates the feedback compensation strategy. To maximize these advancements, we can incorporate the feedforward compensation approach with MEKF. Tables 5 and 6 offers a comparison between the original MEKF (feedback-based) and our refined version of MEKF+Feedforward. As can be seen, the incorporation of feedforward adaptation distinctly enhances the performance of MEKF.

Table 5: Performance (RMSE) comparison between the original MEKF approach and MEKF + feedforward adaptation approach on Robotic-related datasets.

| Method\Dataset | THOR (m) | Assembly (cm) | NGSIM (m) | Robot arm (rad) |
|---|---|---|---|---|
| MEKF | 0.865 ± 0.002 | 1.184 ± 0.001 | 0.895 ± 0.009 | 0.191 ± 0.002 |
| MEKF + Feedforward | **0.831 ± 0.002** | **1.179 ± 0.001** | **0.866 ± 0.009** | **0.179 ± 0.002** |

Table 6: Performance (RMSE) comparison between the original MEKF approach and MEKF + feedforward adaptation approach on general time-series benchmarks.

| Method\Dataset | ETTh1 | Exchange | ILI |
|---|---|---|---|
| MEKF | 0.359 ± 0.008 | 0.595 ± 0.011 | 1.848 ± 0.018 |
| MEKF + Feedforward | **0.346 ± 0.007** | **0.583 ± 0.011** | **1.697 ± 0.018** |

## F.2 Incorporating Feedforward Adaptation with SOTA prediction models

The proposed feedforward adaptation approach can be readily applied to any SOTA prediction model, providing further enhancements.As a demonstration, we incorporated the feedforward adaptation strategy into the Pishgu [39], which has shown promising results on the NGSIM dataset. The results of this application are documented in Table 7.

Table 7: Performance comparison between the original Pishgu model and Pishgu + feedforward adaptation approach on NGSIM dataset.

| RMSE (m) | 1s | 2s | 3s | 4s | 5s |
|---|---|---|---|---|---|
| Pishgu | 0.15 | 0.46 | 0.82 | 1.25 | 1.74 |
| Pishgu + Feedforward Adapt | **0.13** | **0.39** | **0.69** | **1.03** | **1.41** |

## F.3 Computational complexity

The computational complexity of the proposed approach involves two primary steps per iteration: sample selection and optimization using the selected samples. 1) Sample Selection: This step involves selecting samples based on similarity calculations between the current observation and all samples in the buffer. Assume the input sample $X_t \in \mathcal{R}^{I \times m}$ has the input horizon $I$ and the dimension of each horizon (or coordinates) is $m$. $L$ is the size of the buffer containing previous samples. The complexity of sample selection can be estimated as $\mathcal{O}(I^2 \cdot m^2 \cdot L)$. 2) Optimization: The computational complexity of the optimization step is influenced by factors such as the model size and the optimizer used. In our experiments, we utilized a relatively small MLP model with approximately $7.8K$ parameters. Given that the dimensionality of the trajectory dataset is not as extensive as that of images and considering our use of a reasonably sized buffer (e.g., $L = 1000$), the computational load remains manageable. To offer a practical perspective, we present time usage data for various methods in Table 8. It's worth noting that the time required for sample selection

is generally less than that for optimization, further demonstrating the feasibility of our approach in terms of computational efficiency.

Table 8: Time usage (sample selection and optimization) per iteration of different methods (unit: ms) on THOR dataset.

| Time (ms) | ER | A-GEM | SMCD | Feedback | Feedforward |
|---|---|---|---|---|---|
| Sampling | 0.60 | 0.59 | 0.35 | 0.35 | 0.99 |
| Optimization | 2.42 | 4.16 | 2.92 | 2.36 | 2.38 |
| Total | 3.02 | 4.75 | 3.27 | 2.71 | 3.37 |

# G  Further Related Works

## G.1  Related Works in Continual Learning

The proposed feedforward adaptation strategy maintains a memory buffer by storing recent $L$-step observations. It then selects important samples (those with higher similarity to the current sample in our implementation) from the buffer to enhance learning. This method shares similarities with replay-based continual learning, as both approaches involve retaining crucial past samples in a buffer and replaying them to enhance the learning process. However, numerous replay-based continual learning methods, such as Experience Replay [18] and Average Gradient Episodic Memory (A-GEM) [19], rely on random replays. On the contrary, replay-based continual learning methods that employ importance sampling (as opposed to random replay) are specifically designed for scenarios involving task-incremental or class-incremental learning [17, 40, 20]. These methods are often developed within the context of classification tasks, which might not directly align with the task structure of our online adaptation and regression tasks. For instance, methods like iCaRL [41] primarily operate on class-wise samples, and approaches such as Ring Buffer and k-Means based sampling [18] rely on class information for aggregating samples. Our proposed methods do not rely on class or task information to sample important samples based on similarity. Furthermore, online adaptation places its emphasis on the localized performance of multiple upcoming predictions, which underscores the significance of both recalling and forgetting. Conversely, many continual learning methods prioritize a more global and generalized prediction capability, which tends to emphasize the importance of effective recalling.

## G.2  Related Works in Adaptation of Control tasks

Many researchers investigate the online adaptation property of prediction models or policy models in control tasks, aiming to generalize to subtle variations of the environment. As an example, SMCT is a simple and effective approach to adapting neural models in response to changing environments [36]. While both our method and SMCT aim at adapting neural models to changing data distributions, there are notable differences in their approaches. SMCT primarily utilizes feedback data compensation strategies to adapt the dropout layer using particle filters, while our proposed method employs feedforward compensation strategies to extract more crucial samples. Furthermore, SMCT focuses on adapting to dropout layers using gradient-free methods, whereas our approach adapts to arbitrary layers through gradient-based optimization. Latent variable models [42] are oriented towards control problems where predicted outputs impact the environment and subsequent observations. Which is designed to collect many data points or trajectories while interacting with the environment under specific latent variables or dynamics ($e_k$), which is a setup not applicable to our online adaptation of time-series prediction tasks. Meta-learning approach [43] in an online learning context employs strategies like SGD for parameter updates and feedback compensation for sample selection. In contrast, our approach employs feedforward compensation for sample selection and subsequently utilizes these selected samples for model updates.

