# OpenReview forum: "Online Model Adaptation with Feedforward Compensation"
_robot-learning.org/CoRL/2023/Conference — CoRL 2023 Poster_

### Official Review · Reviewer_yQR8 · 2023-07-19

**Confidence:** 4
**Originality:** Good
**Technical Quality:** Good
**Clarity Of Presentation:** Good
**Impact:** 3

**Recommendation:**

Weak Reject: I recommend rejecting the paper, but will not argue for my recommendation if the majority of other reviewers have a different opinion.

**Review:**

The paper addresses an important question in the robotics community: how to efficiently adapt predictive models online based on observed data? The different experiments show that the proposed method has the potential to scale and be applied to different, complex domains. I also appreciate the inclusion of a clear criterion and procedure for applying the proposed framework. Overall, the paper is well-written and the chosen experiments do a good job of showcasing the performance and applicability of the method.

The main weaknesses of the paper relate to the positioning w.r.t the state-of-the-art (i.e., related work) and experimental results.



**Quality Of The Limitations Section:**

Additional details required

**Questions For Rebuttal:**

In terms of the related work, I believe the paper requires a more thorough discussion and comparison with other online adaptation methods (e.g. [1]) and approaches in which past observations are taken into account during adaption via latent representations (e.g. [2]).

In terms of the experimental results, a more detailed description of the experimental setup and observed results is needed. For instance, it will be beneficial to have more details about:

- the buffer size and update process, the number of critical samples required, the dimensionality of datasets/tasks, the meaning of the prediction output, observation and prediction lengths, and critical sample selection. Ablations that show how performance changes based on these choices should also be included
- do metrics reported in Table 1 and the subsequent figures include multiple runs with different seeds or data splits? Although Table 1 shows that the feedforward framework performs better, including results from multiple results (mean + standard deviation) will help the reader to better gauge the overall performance of the proposed method.
- in Figure 1(b.2), why does the feedforward approach result in what looks like a poor prediction output

I also believe that additional comparisons with approaches that capture or use past observations for adaption ([2]) or meta-learning-based models ([3]) would provide insights into the applicability and benefits of the proposed framework.

[1] Carreno, Pamela, Dana Kulic, and Michael Burke. "Adapting Neural Models with Sequential Monte Carlo Dropout." Conference on Robot Learning. PMLR, 2023.

[2] Perez, Christian F., Felipe Petroski Such, and Theofanis Karaletsos. "Efficient transfer learning and online adaptation with latent variable models for continuous control." arXiv preprint arXiv:1812.03399 (2018).

[3] Nagabandi, Anusha, Chelsea Finn, and Sergey Levine. "Deep online learning via meta-learning: Continual adaptation for model-based rl." arXiv preprint arXiv:1812.07671 (2018).

**Robotics Focus:**

Highly relevant to robotics but no hardware experiments

**Summary Of Paper:**

The ability to adapt prediction models online is key for a successful robot deployment in dynamic environments. Existing approaches rely on adaptation schemes where only the latest observation and prediction error are considered, thus making them susceptible to forgetting past, and likely, useful information. In this work, the authors tackle the "forgetting" problem by proposing a feedforward adaptation framework in which past, critical samples stored in a memory buffer are used to adapt the model and generate predictions about the future. The authors also show how the proposed framework provides uncertainty estimates and guarantees on performance when applied to slow time-varying systems. Evaluations and analyses on a toy example and 4 publicly available datasets are also included, showing that the proposed feedforward adaptation framework outperforms the chosen baselines.

**Summary Of Recommendation:**

In the current state, I recommend a “Weak Reject” for this paper. The paper tackles an interesting problem and proposes a novel framework for online model adaption. However, additional clarity and comparisons are required to fully understand the applicability and limitations of the proposed approach.

---

### Official Review · Reviewer_yPZ4 · 2023-07-19

**Confidence:** 3
**Originality:** Good
**Technical Quality:** Fair
**Clarity Of Presentation:** Very Good
**Impact:** 3

**Recommendation:**

Weak Reject: I recommend rejecting the paper, but will not argue for my recommendation if the majority of other reviewers have a different opinion.

**Review:**

The authors propose a method to address prediction/forecasting of (slow) time-varying, dynamics systems. The problem domain is one that is germane to robotics–especially practical applications thereof. The presentation is clear and the theoretical analysis appears sound. The derived uncertainty bounds are perhaps the most interesting and most verifiably novel aspect of the paper.  However, the core novelty and impact of the nearest state experience replay sampling and impact is unclear to the reviewer as the authors do not situate their results in the rich context of other continual learning methods or recent state-of-the-art (SOTA) for time series forecasting.

There is a rich literature focused on continual learning (e.g., refer to the recent survey in [1], especially section 4.2, which enumerates numerous experience replay approaches or [2], which is focused on experience replay). The impact of the paper would be greatly solidified if a theoretical or empirical comparison to other experience replay approaches were included.

The value would be even further cemented if the proposed method was compared against SOTAfor times series forecasting (especially for the target of weakly non-stationary dynamic systems).  This includes a comparison to additional datasets that are more prevalent for time series prediction such as ETT [3] or at least a comparison of SOTA for the datasets considered in the papers.  As an example, the recent work in [4] achieved a RMSE of 0.46m @ 2s for NGSIM but under different settings: a 70% train and 10% validation and 20% test split and with a 3-second input producing a 5-second forecasting prediction. A direct comparison of this approach, or whichever approach has achieved SOTA for NGSIM, Thor, and the Assembly tasks would help the reviewer and reader assess the contribution of the proposed method.

The authors situate their work in the field “ online adaptation methods” (e.g., lines 3 and 93) but do not provide any explicit references to prior work in this area.

The analysis of when to use feedback vs feedforward both along with the error bound prediction \hat{K} is important for estimating the error bound.The analysis in the supplementary system is for a trivial network representing a simple problem. It would be instructive to include how determine this parameter (along with K and \delta) to make the proposed method more applicable.

[1] Wang, Liyuan, Xingxing Zhang, Hang Su, and Jun Zhu. "A comprehensive survey of continual learning: Theory, method and application." arXiv preprint arXiv:2302.00487 (2023).

[2] Bagus, Benedikt, and Alexander Gepperth. "An investigation of replay-based approaches for continual learning." In 2021 International Joint Conference on Neural Networks (IJCNN), pp. 1-9. IEEE, 2021.

[4] Haoyi Zhou, Shanghang Zhang, Jieqi Peng, Shuai Zhang, Jianxin Li, Hui Xiong, and Wancai Zhang. Informer: Beyond efficient transformer for long sequence time-series forecasting. In The Thirty-Fifth AAAI Conference on Artificial Intelligence, AAAI 2021, Virtual Conference, volume 35, pages 11106– 11115. AAAI Press, 2021

[3] Alinezhad Noghre, Ghazal, Vinit Katariya, Armin Danesh Pazho, Christopher Neff, and Hamed Tabkhi. "Pishgu: Universal Path Prediction Network Architecture for Real-time Cyber-physical Edge Systems." In Proceedings of the ACM/IEEE 14th International Conference on Cyber-Physical Systems (with CPS-IoT Week 2023), pp. 88-97. 2023.


**Quality Of The Limitations Section:**

Additional details required

**Questions For Rebuttal:**

The authors situate their work in the field “ online adaptation methods” (e.g., lines 3 and 93) but do not provide any explicit references to prior work in this area. Could you add further citations for this area of research?

The analysis of when to use feedback vs feedforward both along with the error bound prediction \hat{K} is important for estimating the error bound.The analysis in the supplementary system is for a trivial network representing a simple problem. It would be instructive to include how determine this parameter (along with K and \delta) to make the proposed method more applicable. Would it be possible to provide such analysis?

Could the authors provide additional experiments for ETT and/or compare SOTA results for the datasets they tested including NGSIM, Thor, and Assembly?

**Robotics Focus:**

Sufficient demonstration on hardware

**Summary Of Paper:**

To address time-varying-drift for time series (or sequential) forecasting problems, the authors of this work present a method called “Online Adaptation with Feedforward Compensation.” This approach continuously updates a pre-trained model by preferentially sampling the states closest to the current state from an experience replay buffer. The authors provide theoretical analysis of this approach’s benefits and trade-offs relative to a naive sampling of the latest state’s experience buffer, which they term Feedback Adaptation. They further present a method to bound prediction uncertainty.

Empirical results are provided for a toy example, which is used to explore the benefits and trade-offs of the method based on the degree of non-stationarity and the experience replay buffer length. The paper presents results comparing variations in sampling of their approach to several robotic specific time series forecasting data sets along with a real world pick and place manipulation task.

**Summary Of Recommendation:**

The proposed method is clear and relevant to the field of robotics. While the paper provides theoretical analysis to ground the proposed approach compared to trivial alternatives, it lacks a thorough comparison to state of the art time series forecasting techniques and other experience replay sampling methods for continual learning. The reviewer is unable to fully assess the impact of the proposed method because of this lack of evidence. As it stands, the reviewer recommends the paper be rejected unless these shortcomings are addressed.

---

### Official Review · Reviewer_GpzZ · 2023-07-19

**Confidence:** 3
**Originality:** Poor
**Technical Quality:** Poor
**Clarity Of Presentation:** Very Good
**Impact:** 2

**Recommendation:**

Weak Reject: I recommend rejecting the paper, but will not argue for my recommendation if the majority of other reviewers have a different opinion.

**Review:**

- Strength
  - Paper is well written and easy to follow.
  - The topic, online adaptation for neural network, is crucial
- Weakness
  - Need more comparison with existing work. In my understanding, proposed method can be categorized as continuous learning with replay buffer, which has abundant existing works to be compared with.
  - As stated in the limitation, the method relies on an assumption that there is not abrupt change in data distribution, which is not true. Imagine going pass a cross: trajectory distribution before the cross and after the cross is significantly different. Thus the performance of the proposed method is questionable. The benefit may solely come from the assumption that there is no abrupt change in the data distribution (which is not true as stated above), instead of really making the model learn.
  - Questionable dataset/benchmark selection. Selected dataset is **not** widely used for benchmarking trajectory prediction. For example, THOR is only cited by 43 other work, Assembly is cited by 21 other works, NGSIM is only cited by 4, robotic arm is collected by authors. Given that there are abundant datasets for trajectory prediction, this makes the evaluation results less reliable.

**Quality Of The Limitations Section:**

Additional details required

**Questions For Rebuttal:**

- Why the proposed method is called "feedforward"? The proposed method is actually still using error back-propagation like "feedbackward".

- What's the difference between the proposed method to existing works in the continuous learning area?

- What's the performance on more complex tasks?

**Robotics Focus:**

Sufficient demonstration on hardware

**Summary Of Paper:**

This paper proposes a continuous online learning method for trajectory prediction. For every input, before preforming inference, proposed method will look for the most similar data sample in a buffer and finetune the model with the sample.

**Summary Of Recommendation:**

Given that paper:
- Lacks results against existing baseline methods,
- Relies on not valid assumptions
- Has questionable evaluation results

a weak rejection is given.

---

### Official Review · Reviewer_sMeX · 2023-07-20

**Confidence:** 4
**Originality:** Very Good
**Technical Quality:** Very Good
**Clarity Of Presentation:** Excellent
**Impact:** 4

**Recommendation:**

Strong Accept: I recommend accepting the paper and will argue for my recommendation even if other reviewers hold a different opinion.

**Review:**

quality: It is a high quality paper and I enjoyed reading it

clarity: The paper is well written and the presented method is clear

originality: The idea seems to be original and, as far as I know, a new and interesting way to see the problem of online model adaptation

significance: As the general pipeline is not new but just with other ML models, the significants of this work might be limited

Strengths:
- The paper presents an interesting way to think about online model adaptation, i.e., from the perspective of feedback and feedforward control
- The proposed approach seems to be fairly simple but outperforms other standard methods

Weaknesses:

- I think that the paper could be strengthen by connecting the presented idea with other related work. For instance, there is a lot of work about information gain of data, i.e., which data is most representative or other works, e.g., “Robust Online Model Adaptation by Extended Kalman Filter”


**Quality Of The Limitations Section:**

Limitations are addressed clearly

**Questions For Rebuttal:**

- What is the computational complexity of the proposed approach?
- How does it related to the idea of having local models, for instance, local GP models where, depending on the current state, the closest model is activated and used for predictions. Instead of having a memory, can it be seen as some “local” models?

**Robotics Focus:**

Highly relevant to robotics but no hardware experiments

**Summary Of Paper:**

In this paper, the authors present an online model adaptation approach. In contrast to many existing approaches where simply the “oldest” data will be discarded, the proposed approach works more in an feedforward fashion, i.e., it uses critical data samples from a memory buffer instead the latest sample. Furthermore, under some mild assumptions, error bound for the prediction are presented. A toy example and experiments with real world data sets show the superior of the proposed approach.

**Summary Of Recommendation:**

I really enjoyed reading the paper and the feedback/feedforward perspective on online model adaptation is interesting and worth to publish.

---

### Author Response · Authors · 2023-08-10
**Revision of the submission**

We wish to express our heartfelt appreciation to the reviewers for their invaluable insights. In response to the feedback provided, we have revised our manuscript.   Within the manuscript, we have distinguished the revised sections by employing colored.
**In the revision, we conducted an extended comparative study and expanded our experimentation with additional datasets.**

**Datasets.** In addition to the evaluation of four robotics-related datasets presented in the initial version of this paper, we further assess the proposed feedforward adaptation method using three real-world time-series benchmarks: ETTh1 (Electricity Transformer Temperature hourly-level dataset) [1], Exchange-Rate [2], and ILI (Influenza-like Illness) dataset [3].

**Baselines.** 1) *w/o adapt* performs prediction without any adaptation.
2) *Feedback adaptation* utilizes the latest observations to adapt its models  [5].  3) *Experience Replay (ER)* introduces random sample replay from this buffer to enhance learning [6] . 4)  *Average Gradient Episodic Memory (A-GEM)* doesn't directly minimize loss on replayed samples. Instead, it aims to minimize loss on current data under the constraint of avoiding loss increase on replayed data [7]. 5) *Sequential Monte Carlo Dropout (SMCD)* employs a particle filter to sustain a distribution over dropout masks, thereby dynamically adapting the neural model to evolving settings [8].

**Results.** For every experimental configuration, we conducted experiments across 10 distinct random seeds and present the mean results alongside their corresponding standard deviations. The evaluation metric employed is the root-mean-squared error (RMSE) for prediction outcomes. Results pertaining to the robotics-related datasets are tabulated in Table 1. Evidently, the proposed feedforward adaptation method consistently showcases superior performance across all four datasets. Additionally, results for the general time-series prediction benchmarks are presented in Table 2. Notably, the feedforward adaptation method maintains its performance lead, surpassing other baselines, even including the continual learning strategies ER and A-GEM.

Table 1. Performance (RMSE) comparison between the proposed feedforward adaptation method and other baselines on Robotic-related datasets.
| Method\Dataset |     THOR     |   Assembly  |    NGSIM    |  Robot arm  |    ETTh1    |   Exchange  |     ILI     |
|:--------------:|:------------:|:-----------:|:-----------:|:-----------:|:-----------:|:-----------:|:-----------:|
|    w/o adapt   |  1.208±0.005 | 1.324±0.001 | 1.203±0.011 | 0.257±0.001 | 0.195±0.006 | 0.549±0.004 | 4.348±0.010 |
|       ER       |  0.914±0.002 | 1.194±0.001 | 0.985±0.015 | 0.216±0.001 | 0.162±0.007 | 0.349±0.007 | 3.869±0.020 |
|      A-GEM     |  0.873±0.002 | 1.188±0.001 | 0.935±0.012 | 0.210±0.001 | 0.148±0.007 | 0.337±0.008 | 3.752±0.016 |
|      SMCD      |  0.937±0.006 | 1.201±0.001 | 1.002±0.015 | 0.210±0.002 | 0.171±0.008 | 0.402±0.004 | 4.001±0.010 |
|    Feedback    |  0.891±0.002 | 1.191±0.001 | 0.963±0.012 | 0.214±0.001 | 0.153±0.006 | 0.349±0.008 | 3.868±0.016 |
|   Feedforward  | 0.839 ±0.002 | 1.180±0.001 | 0.890±0.011 | 0.193±0.001 | 0.128±0.006 | 0.311±0.008 | 3.041±0.014 |

Table 2. Performance (RMSE) comparison between the proposed feedforward adaptation method and other baselines on general time-series benchmark.
| Method\Dataset |     ETTh1     |    Exchange   |      ILI      |
|:--------------:|:-------------:|:-------------:|:-------------:|
|    w/o adapt   | 0.485 ± 0.011 | 0.783 ± 0.006 | 2.195 ± 0.009 |
|       ER       | 0.391 ± 0.013 | 0.601 ± 0.013 | 1.943 ± 0.016 |
|      A-GEM     | 0.373 ± 0.013 | 0.619 ± 0.013 | 1.906 ± 0.012 |
|      SMCD      | 0.413 ± 0.015 | 0.673 ± 0.008 | 2.051 ± 0.010 |
|    Feedback    | 0.383 ± 0.013 | 0.619 ± 0.012 | 1.953 ± 0.013 |
|   Feedforward  | 0.357 ± 0.012 | 0.589 ± 0.012 | 1.843 ± 0.011 |

Please refer to the details provided in Section 5 of the revised manuscript for more information.



[1] H. Zhou, et al. "Informer: Beyond efficient transformer for long sequence time-series forecasting". AAAI, 2021.
[2] G. Lai, et al. " Modeling long-and short-term temporal patterns with deep neural networks". ACM SIGIR, 2018.
[3] ILI dataset website: \url{https://gis.cdc.gov/grasp/fluview/fluportaldashboard.html}
[4] A. Zeng, et al. "Are transformers effective for time series forecasting?." AAAI, 2023.
[5] A. Abuduweili et al. "Robust nonlinear adaptation algorithms for multitask prediction networks". International Journal of Adaptive Control and Signal Processing,  2021
[6] A. Chaudhry, et al. "On tiny episodic memories in continual learning". arXiv preprint, 2019
[7] A. Chaudhry, et al, "Efficient lifelong learning with a-gem". ICLR, 2019
[8] P. Carreno, et al. "Adapting Neural Models with Sequential Monte Carlo Dropout." CoRL, 2023.

---

### Decision · Program_Chairs · 2023-08-30

**Decision:**

Accept (Poster)

**Comment:**

The paper proposes a method for online model adaptation using past experiences, adapting the model prior to predicting the next data point using this historic data, hence, the term of feedforward adaptation is introduced.
Overall, the reviewers agreed on the importance of the presented problem, the interesting approach, and the very good presentation. But they also agreed on the main weakness of the paper in its current form, which is the comparison to state-of-the-art methods – both in terms of theoretical embeddings as well as in the conducted experiments. During the rebuttal, the authors provided extensive new and requested evaluations and clarifications. The updated version is a great improvement, grounds the interesting and promising idea onto a more solid foundation, and is a valuable contribution to CoRL.

**Importantly though, the updated paper is too long and the authors need to shorten the paper to the required 8 pages without removing important and significant content!**